# Mapping the Scientific Landscape of Bacterial Influence on Oral Cancer: A Bibliometric Analysis of the Last Decade's Medical Progress

**Suh-Woan Hu** [1,2], **Jaw-Ji Yang** [1] **and Yuh-Yih Lin** [1,2,3,*]

1   Institute of Oral Sciences, College of Oral Medicine, Chung Shan Medical University, Taichung 40201, Taiwan; suhwoan@csmu.edu.tw (S.-W.H.); jjyang@csmu.edu.tw (J.-J.Y.)
2   Department of Stomatology, Chung Shan Medical University Hospital, Taichung 40201, Taiwan
3   School of Dentistry, College of Oral Medicine, Chung Shan Medical University, Taichung 40201, Taiwan
*   Correspondence: yuhyih@csmu.edu.tw; Tel.: +886-4-2471-8668 (ext. 55515)

**Abstract:** The research domain investigating bacterial factors in the development of oral cancer from January 2013 to December 2022 was examined with a bibliometric analysis. A bibliometric analysis is a mathematical and statistical method used to examine extensive datasets. It assesses the connections between prolific authors, journals, institutions, and countries while also identifying commonly used keywords. A comprehensive search strategy identified 167 relevant articles, revealing a progressive increase in publications and citations over time. China and the United States were the leading countries in research productivity, while Harvard University and the University of Helsinki were prominent affiliations. Prolific authors such as Nezar Al-Hebshi, Tsute Chen, and Yaping Pan were identified. The analysis also highlights the contributions of different journals and identifies the top 10 most cited articles in the field, all of which focus primarily on molecular research. The article of the highest citation explored the role of a *Fusobacterium nucleatum* surface protein in tumor immune evasion. Other top-cited articles investigated the correlation between the oral bacteriome and cancer using 16S rRNA amplicon sequencing, showing microbial shifts associated with oral cancer development. The functional prediction analysis used by recent studies has further revealed an inflammatory bacteriome associated with carcinogenesis. Furthermore, a keyword analysis reveals four distinct research themes: cancer mechanisms, periodontitis and microbiome, inflammation and *Fusobacterium*, and risk factors. This analysis provides an objective assessment of the research landscape, offers valuable information, and serves as a resource for researchers to advance knowledge and collaboration in the search for the influence of bacteria on the prevention, diagnosis, and treatment of oral cancer.

**Keywords:** bacteria; mouth neoplasms; bibliometrics; databases; bibliographic

## 1. Introduction

Oral cancer has a significant impact on public health, including morbidity and mortality rates, as well as socioeconomic burdens [1]. The latest survey by the National Cancer Institute of the United States (2015–2019) indicated that 11.5 adults per 100,000 will develop oral cancer, which represents 2.8% of all new cancer cases [2,3]. Left untreated, oral cancer can spread throughout the mouth to other areas of the head and neck. The distant metastasis of oral cancer may involve other major organs, such as the lungs [4]. Despite advances in surgical techniques, adjuvant radiotherapy, and chemotherapy, it was estimated that about 31.5% of patients with oral cancer could not survive for more than 5 years after diagnosis, even after being actively treated [5]. Consequently, the increasing number of oral cancer cases worldwide, as predicted by the World Health Organization, requires urgent attention as a significant public health issue.

Oral carcinogenesis is a multifactorial process that involves many risk factors [6], such as drinking, smoking, chewing betel, poor oral hygiene, and poor eating habits. Other possible risk factors [7] can include viral infection, fungal infection, and chronic periodontitis. Currently, the effects of the oral microbiota are intensively studied in association with the onset and progression of oral cancer. Bacteria contribute the most to the composition of the oral microbiota, and fungi and viruses constitute a smaller proportion. Dysbiosis, characterized by changes in microbial homeostasis by reducing bacterial diversity and, therefore, resulting in an increased representation of pathogenic microorganisms, has been associated with the development of oral cancer [8,9]. Studies have also highlighted the presence of a group of specific microbiota, such as *Fusobacterium*, *Prevotella*, *Porphyromonas*, and *Lactobacillus*, within tumor tissues, suggesting its potential involvement in cancer progression [10,11].

Early detection of oral cancer is crucial to achieving favorable clinical outcomes and, therefore, higher survival rates [12]. The association between oral cancer and bacteria has not been clearly investigated until now, and the possible underlying mechanisms remain a significant research challenge. Consequently, research on bacteria-associated oral cancer has gained substantial attention, leading to an increasing number of related publications [13–16]. Understanding the possible bacterial association with oral cancers may have a great impact on the diagnosis and treatment of this particular disease.

Bibliometric analysis [17] objectively assesses past large data to quantify the impact of a study in its specific field of science [18]. To date, several studies have explored the association between oral cancer and bacteria [19–21]; however, to our knowledge, no bibliometric analysis has been performed on this topic. In this study, we performed a comprehensive bibliometric analysis focusing on the literature published in the last decade. The analysis included article citations, country of origin, publishing journals, authors and their affiliations, highly cited studies, and keywords. In addition, we examined the research trends and their hotspots. This analysis aims to provide researchers with an objective understanding of the research situation in this field and to serve as a reference for further in-depth investigations.

## 2. Materials and Methods

### 2.1. Data Sources and Search Strategies

Research on oral cancer and bacteria from January 2013 to December 2022 was carried out using the Thomson Reuters Web of Science database (WoS). This database was chosen due to its extensive collection of medical literature and comprehensive citation analysis [22]. The Science Citation Index Expanded (SCIE) and Social Science Citation Index (SSCI) databases were utilized. The searches were performed on 22 May 2023 to ensure the avoidance of database renewal bias. The search strategy (Figure 1) included specific terms related to cancer (e.g., carcinoma, tumor) and the oral region (e.g., oral, mouth). Additionally, terms related to bacteria and the microbiota were included while excluding viral, fungal, and mycotic terms.

| 1: | TS = (cancer* *OR carcino** OR tumor* *OR tumour** OR malign* *OR neoplasm**)<br>1,381,822 studies identified |
|---|---|
| 2: | TS = ((oral OR intraoral OR mouth OR tongue OR buccal OR lip OR gingiva *OR palate)*<br>432,990 studies identified |
| 3: | TS = ((bacteri* *OR microbiota* OR microbiome *OR microflora* OR microorganism)*<br>*NOT (vir** OR *myco** OR *fung**))<br>908,372 studies identified |
| 4: | #1 AND #2 AND #3<br>1886 studies identified |

**Figure 1.** Search strategy in WOS (performed on 22 May 2023). *wildcard characters. #searching step.

## 2.2. Data Collection and Processing

Only original articles written in English and published between 2013 and 2022 were included in this study (Figure 2). Since our goal was to find research on bacteria-associated oral cancer, any irrelevant article was manually screened by two authors (S.-W. Hu and Y.-Y. Lin) through abstracts and full texts. In case of discrepancy, the articles in question were kept in the study. After the process of exclusion, the bibliometric data of the included references were exported from WoS for further investigation. Data on the authors, journal, affiliation, publication date, country/region of origin, number of citations, and H-index were extracted from the identified publications.

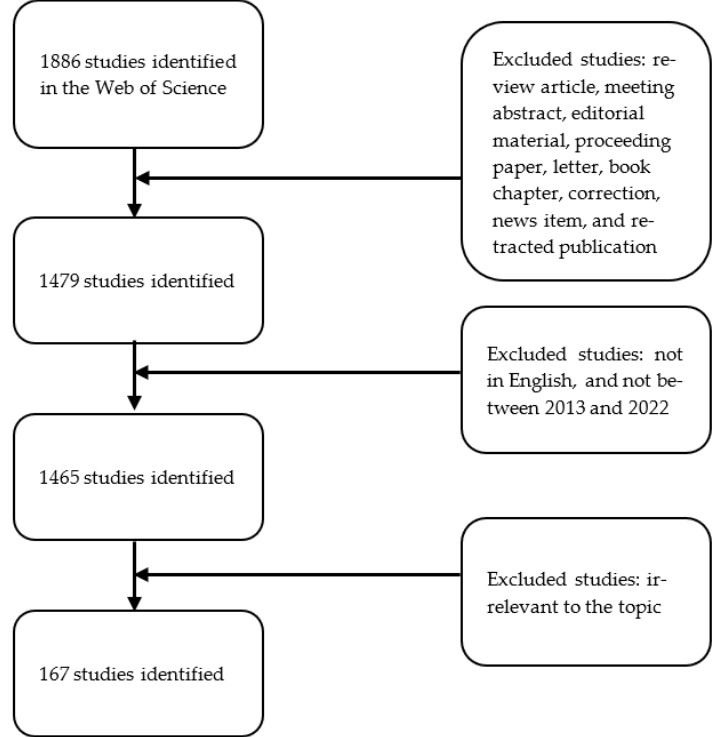

**Figure 2.** Flow diagram of the inclusion process. Articles irrelevant to the topic "oral cancer and bacteria" were excluded (manually screened by authors through abstracts and full texts).

*2.3. Bibliometric Analysis*

The WoS covers extensive biomedical research and contains tools for citation analysis. The H-index [23], which evaluates the academic productivity of sources, such as author, affiliation, and country/region, was evaluated. The impact factor (IF), obtained from the journal citation reports, was used as an indicator to evaluate the journals. Both objective indexes (e.g., number of publications, authors, affiliations, citation counts, country/region) and subjective indexes (e.g., keywords) were recorded for descriptive bibliometric analysis. Version 1.6.19 of VOSviewer was used to generate bibliometric network maps for co-authorship by country/region and for keywords [24].

## 3. Results

*3.1. Publications and Their Citations*

Between 2013 and 2022, a total of 167 articles on the topic of oral cancer and bacteria met the search criteria and were included in the evaluation. The number of publications showed a progressive increase over time, with the highest number of publications observed in 2022, reaching 35 articles (Figure 3). The citations of these articles also showed an upward trend, indicating the growing impact and influence of research in this area. By 2022, the total number of citations had risen significantly to 1350.

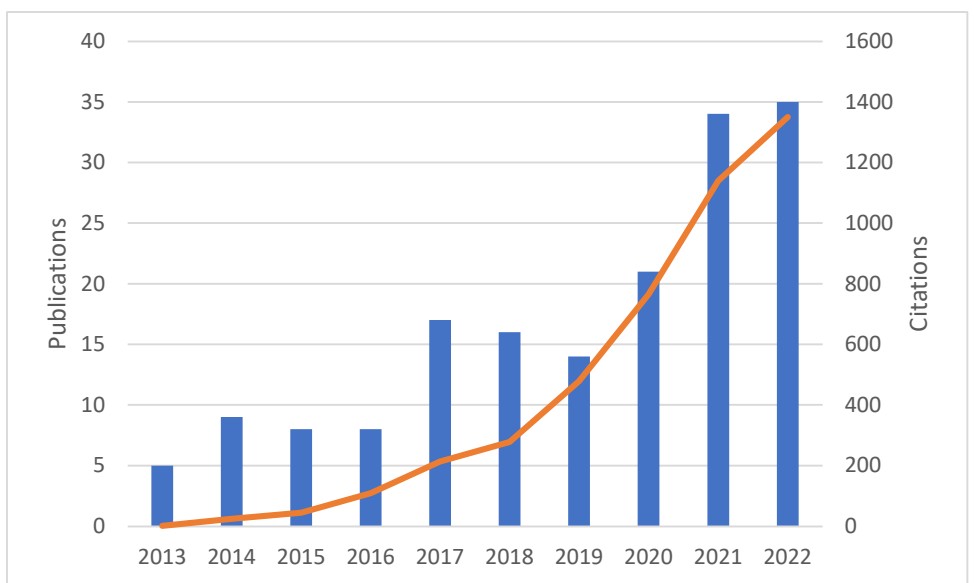

**Figure 3.** Publications (bar) and their citations (line) over time.

*3.2. Authors, Affiliations, and Countries/Regions*

The authorship productivity and citation counts were assessed to identify the most prolific authors (Table 1). Notable authors with significant contributions include Nezar Al-Hebshi (6 publications, 342 citations), Tsute Chen (5 publications, 368 citations), Yaping Pan (5 publications, 177 citations), Caj Haglund (5 publications, 98 citations), and Jaana Hagström (5 publications, 98 citations). These authors have made substantial research contributions, as evidenced by their publication output and impact on citations. In terms of the H-index ranking, which considers both publication output and citation impact, Al-Hebshi, Chen, and Pan were tied with H-index 5, indicating their influential presence in the field.

**Table 1.** The most productive authors and their affiliations.

| Authors | Publications | Affiliation | Citations | H-Index |
|---|---|---|---|---|
| Al-Hebshi NN [25–30] | 6 | Temple Univ, USA; Jazan Univ, Saudi Arabia | 342 | 5 |
| Chen T [27–31] | 5 | Harvard Univ, USA; Forsyth Institute, USA | 368 | 5 |
| Pan YP [32–36] | 5 | China Medical Univ, China | 177 | 5 |
| Haglund C [37–41] | 5 | University of Helsinki, Finland | 98 | 4 |
| Hagstrom J [37–41] | 5 | University of Helsinki, Finland | 98 | 4 |

In terms of affiliations, Harvard University demonstrated the highest level of contribution with 12 publications (Table 2). The University of Helsinki and Sichuan University followed closely, both with 10 publications each. The Pennsylvania Commonwealth System of Higher Education (PCSHE) and the State University System of Florida both had seven publications each. The prominent presence of Harvard University in the field is reflected in the substantial impact of citations, with a total of 1207 citations and an H-index of 11. The University of Helsinki, located in Finland, also showcased significant contributions, accumulating 212 citations and an H-index of 9.

**Table 2.** Most productive affiliations and their countries.

| Affiliations | Publications | Citations | H-Index |
|---|---|---|---|
| Harvard University, USA | 12 | 1207 | 11 |
| University of Helsinki, Finland | 10 | 212 | 9 |
| Sichuan University, China | 10 | 91 | 5 |
| Pennsylvania Commonwealth System of Higher Education PCSHE, USA | 7 | 299 | 6 |
| State University System of Florida, USA | 7 | 218 | 7 |
| Forsyth Institute, USA | 6 | 368 | 5 |
| China Medical University, China | 6 | 193 | 6 |
| Helsinki University Central Hospital, Finland | 6 | 109 | 5 |
| Karolinska Institutet, Finland | 6 | 113 | 5 |

A country's productivity in the field of oral cancer and bacteria was assessed (Figure 4). Analysis of the data presented revealed the main contributors in terms of the total number of publications. China emerged as the leading country with 56 publications, which is 33.53% of the total. The United States followed closely with 53 publications, accounting for 31.74% of the research output. India produced 14 publications (8.38%). Finland, Japan, and Taiwan also made significant contributions, with 10 publications each (5.99%). In terms of citation impact, the United States exhibited the highest cumulative citation count of 2495, highlighting the influential nature of its research contributions in the field of oral cancer and bacteria. China ranked second with 1294 citations. Furthermore, the United States secured the first position with an H-index of 24, underscoring its significant scientific output and impact on citations. China ranked second with an H-index of 17.

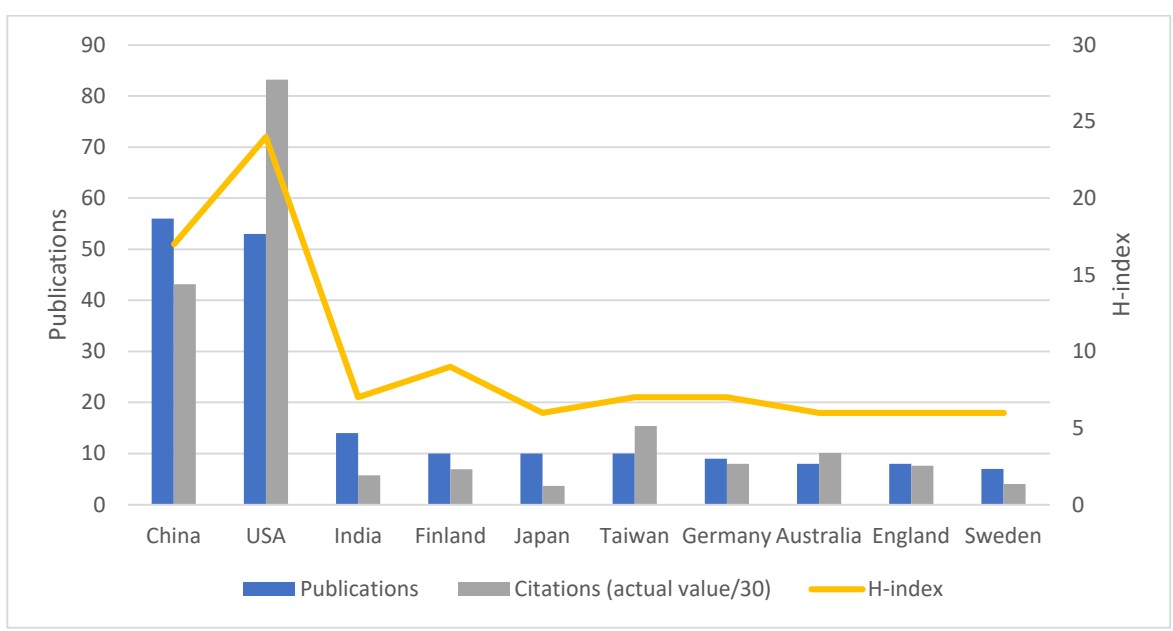

**Figure 4.** Contributions of different countries/regions to research on oral cancer and bacteria.

The co-authorship network of countries is depicted in Figure 5. This visualization of the network provides information on collaborative research efforts between various regions, such as China and the United States, in the study of oral cancer and bacteria.

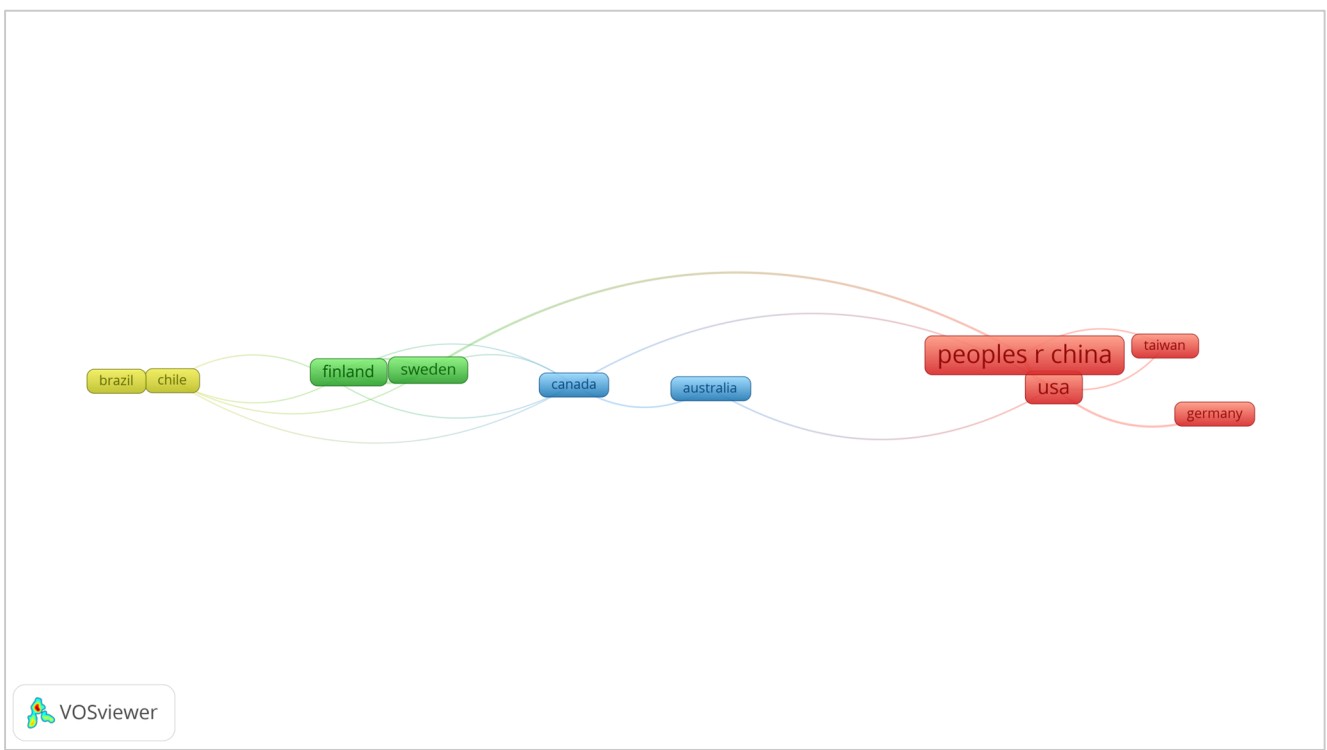

**Figure 5.** Co-authorship network of different countries/regions.

### 3.3. Journals

The contributions of different journals to the field of interest were examined (Table 3). *Frontiers in Cellular and Infection Microbiology*, with an impact factor (IF) of 6.073, published the highest number of studies, accounting for 10 publications (5.99%). *Scientific Reports* ranked second with nine publications (5.39%), and *Frontiers in Microbiology* ranked

third with seven publications (4.19%). It is worth noting that three of the top 10 journals, *Frontiers in Cellular and Infection Microbiology*, *Frontiers in Microbiology*, and *Frontiers in Oncology*, were published by the same publisher, Frontiers Media SA. This highlights the significant contribution of this publisher to this field of research.

**Table 3.** Most productive journals.

| Journal | IF | Publisher | Publications | Average Cites | Scope |
|---|---|---|---|---|---|
| *Frontiers in Cellular and Infection Microbiology* | 6.073 | Frontiers Media SA | 10 | 18.1 | Microbiology |
| *Scientific Reports* | 4.997 | Nature Portfolio | 9 | 58.0 | Multidisciplinary Sciences |
| *Frontiers in Microbiology* | 6.064 | Frontiers Media SA | 7 | 47.1 | Microbiology |
| *Journal of Oral Microbiology* | 5.833 | Taylor & Francis Ltd | 6 | 23.7 | Microbiology |
| *Oncotarget* | 5.168 | Impact Journals LLC | 5 | 101.0 | Cancer |
| *Journal of Dental Research* | 8.924 | Sage Publications Inc | 4 | 46.3 | Dentistry |
| *Oral Oncology* | 5.972 | Elsevier | 4 | 26.3 | Dentistry, Oncology |
| *Frontiers in Oncology* | 5.738 | Frontiers Media SA | 4 | 10.8 | Cancer |
| *International Journal of Molecular Sciences* | 6.208 | MDPI | 4 | 5.5 | Biochemistry and Molecular Biology |
| *Journal of Cancer* | 4.478 | Ivyspring Int Publ | 4 | 2.3 | Cancer |

### 3.4. Top 10 Most Cited Articles

In terms of citation impact, the top 10 most cited articles in the field of oral cancer and bacteria are listed in Table 4. These studies primarily focus on molecular research. The most cited article, authored by Bachrach et al. [25] from Israel and published in *Immunity*, investigated the role of the Fap2 protein in tumor immune evasion. The study shows how *Fusobacterium nucleatum* binds to tumor cell surfaces, facilitating its interaction with the inhibitory receptor TIGIT of NK cells through its Fap2 protein. This interaction shields tumor cells from immune cell attacks, contributing to cancer progression. The findings shed light on the potential mechanisms employed by periodontal pathogens to protect tumors from immune cell attack.

The articles ranked 2 and 4 to 10 explored the correlation between the oral bacteriome and cancer using molecular techniques, particularly 16S rRNA amplicon sequencing. However, these studies show inconsistent results regarding specific species or microbial shifts associated with oral cancer. The limitation of these investigations lies in their narrow focus on compositional analysis. To overcome this limitation, some of the more recent studies [26–28] used functional prediction analysis, which revealed an enrichment of proinflammatory bacterial attributes, indicating the presence of an inflammatory bacteriome.

The third most cited article [29] explored the influence of bacterial infection on cancer development in a mouse carcinogenesis model. The study elucidated how periodontal pathogens interact with cells through Toll-like receptors and the IL-6-STAT3 axis, providing valuable insights into the mechanisms underlying the impact of bacterial infection on cancer progression.

By elucidating these interactions, the research contributes to a deeper understanding of the complex relationship between oral bacteria and cancer, offering potential implications for therapeutic interventions and preventive measures. The findings of this highly cited study emphasize the importance of considering bacterial infections as possible contributing factors to the development and progression of oral cancer. As such, it highlights the importance of exploring novel strategies to target these interactions for better cancer management.

**Table 4.** Top 10 most cited articles.

| Rank | Article Title | Corresponding Author | Year of publication | Country/Region | Journal | Citations | Reference |
|---|---|---|---|---|---|---|---|
| 1 | Binding of the Fap2 protein of *Fusobacterium nucleatum* to human inhibitory receptor TIGIT protects tumors from immune cell attack | Bachrach, Gilad | 2015 | Israel | *Immunity* (IF =32.4) | 646 | [42] |
| 2 | Changes in abundance of oral microbiota associated with oral cancer | Albertson, Donna G. | 2014 | USA | *PLOS one* (IF = 3.7) | 212 | [15] |
| 3 | Periodontal pathogens *Porphyromonas gingivalis* and *Fusobacterium nucleatum* promote tumor progression in an oral-specific chemical carcinogenesis model | Elkin, Michael | 2015 | Israel | *Oncotarget* (IF = 5.168) | 208 | [43] |
| 4 | 16S rRNA amplicon sequencing identifies microbiota associated with oral cancer, human papilloma virus infection, and surgical treatment | Guerrero-Preston, Rafael; Sidransky, David | 2016 | USA | *Oncotarget* (IF = 5.168) | 178 | [44] |
| 5 | Variations in oral microbiota associated with oral cancer | Liang, Jingping | 2017 | China | *Scientific Reports* (IF = 4.6) | 164 | [45] |
| 6 | Oral microbiota community dynamics associated with oral squamous cell carcinoma staging | Chang, Kai-Ping | 2018 | Taiwan | *Frontiers in Microbiology* (IF = 5.2) | 155 | [46] |
| 7 | Association of oral microbiome with risk for incident head and neck squamous cell cancer | Hayes, Richard B. | 2018 | USA | *JAMA Oncology* (IF = 28.4) | 150 | [47] |
| 8 | The oral microbiota may have influence on oral cancer | Zhang, Chen Ping | 2020 | China | *Frontiers in Cellular and Infection Microbiology* (IF = 5.7) | 132 | [48] |
| 9 | Inflammatory bacteriome featuring *Fusobacterium nucleatum* and *Pseudomonas aeruginosa* identified in association with oral squamous cell carcinoma | Al-hebshi, Nezar Noor | 2017 | USA, Saudi Arabia | *Scientific Reports* (IF = 4.6) | 126 | [28] |
| 10 | Inflammatory bacteriome and oral squamous cell carcinoma | Al-hebshi, Nezar Noor | 2018 | USA | *Journal of Dental Research* (IF = 7.6) | 96 | [27] |

*3.5. Keyword Analysis of Research Themes in Oral Cancer and Bacteria*

We performed a keyword analysis using VOSviewer (Centre for Science and Technology Studies, Leiden University, The Netherlands) to identify the prominent research themes in the literature related to the role of bacteria in the process of oral cancer development (Figure 6). Four clusters were formed by the identified keywords. The green cluster encompassed studies that focused on the mechanisms underlying cancer development and progression. The keywords included cancer, therapy, apoptosis, migration, invasion, mechanisms, adhesion, proliferation, and activation. The red cluster represented studies that explored the relationship between periodontitis, the oral microbiome, and oral cancer. Keywords such as microbiome, periodontitis, head, gene growth, smokeless tobacco, health, biomarkers, diversity, and *Streptococcus anginosus* were present in this cluster. The blue cluster comprised studies that focused on inflammation and the role of specific bacteria, particularly *Fusobacterium* and *Porphyromonas*, in oral cancer. The yellow cluster included studies that examined various risk factors associated with oral cancer. A supplementary Figure S1 showing the co-occurrence network of bacterium-associated keywords of relevant publications is also provided.

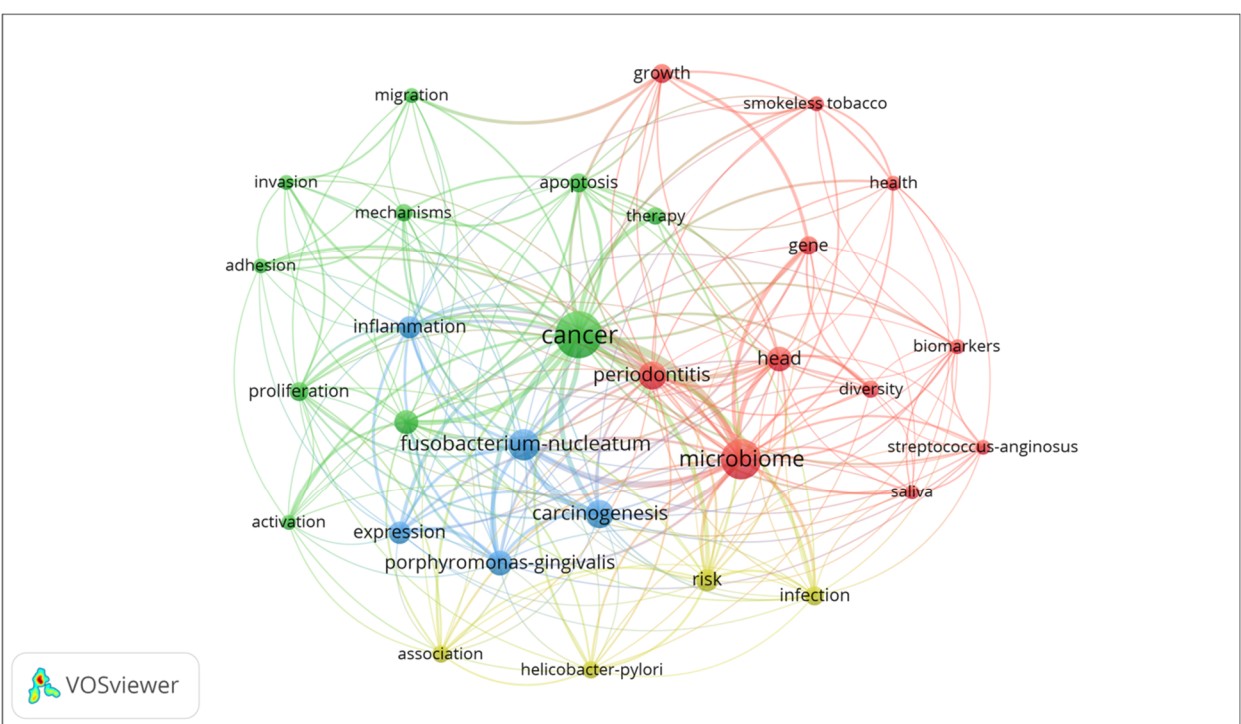

**Figure 6.** Co-occurrence network of keywords of relevant publications.

## 4. Discussion

This study provides a comprehensive evaluation of academic publications that focus on the association between oral cancer and bacteria from 2013 to 2022. The analysis reveals a consistent increase in the number of publications over time, indicating the growing interest in this research area. Moreover, the citations of these articles also demonstrate a positive trend, underscoring their influence and significance within the scientific community. This reflects the growing global severity of oral cancer [49] and the urgent need to understand the role of bacteria in its development [13,50].

Among the authors investigated, Al-Hebshi NN emerges as the most productive, having published six articles exploring the impact of oral bacteria on cancer development, progression, and associated inflammatory processes [25–30]. Al-Hebshi et al. have examined compositional differences in the bacteriome in relation to oral carcinogenicity and have explored the therapeutic potential of health-associated oral bacteria. Additionally, an emphasis is placed on the development of methodologies for bacterial species analysis using advanced techniques such as 16S rRNA sequencing. The application of these techniques has allowed researchers to identify a specific microbiota associated not only with oral cancer but also with factors such as surgical treatment.

Our analysis highlights the substantial contributions made by China and the United States in terms of publication output, citation impact, and overall influence in the field. Additionally, we observe emerging research activities in India, Finland, Japan, and Taiwan, indicating their increasing presence and impact in this domain. Collaborative efforts between these geographic regions have the potential for knowledge exchange and further advancement of research in the field. This academic cooperation is also evident among research institutions, as demonstrated by the co-author network of affiliations analyzed using VOSviewer (see Supplementary Figure S2).

Regarding the journals that have published the most studies in the field, many of them focus on oncology, microbiology, and dentistry. This indicates the multidisciplinary nature of the research conducted in this area, where various fields intersect to contribute to the understanding of the topic.

The most cited articles identified in our study have made significant contributions, addressing various aspects of the topic, including changes in the oral microbiota, the role of specific pathogens, and associations with tumor progression. In particular, the dynamics of the oral microbiota community [46,48] have been one of the main focal points. Investigations have demonstrated that alterations in the composition of the oral microbiota occur at different stages of oral cancer, indicating a potential relationship between microbial shifts and disease progression. Furthermore, variations in the abundance and diversity of microbial communities have been observed and may be associated with the presence and progression of the disease. These findings suggest that bacterial dysbiosis or some specific bacterial species such as *Fusobacterium*, *Prevotella*, *Porphyromonas,* and *Lactobacillus* could potentially serve as biomarkers for assessing cancer risk [11,51].

The influence of bacteria on oral cancer extends beyond its composition to encompass the concept of an inflammatory bacteriome [27,28]. The host's immune response to oral bacteria can lead to chronic inflammation, which has been implicated in the development and progression of oral cancer. Long-term bacterial infections, particularly those involving specific species such as *F. nucleatum* and *P. gingivalis*, have been associated with periodontitis that can contribute to the pathogenesis of oral cancer [52,53].

Through the application of keyword analysis using VOSviewer, our study identified four distinct research themes in the field of oral cancer and bacteria. The green cluster focused on cancer mechanisms, delving into the underlying cellular and molecular processes involved in oral cancer development. This theme sheds light on key factors and pathways that contribute to tumorigenesis. The red cluster explored the intricate relationship between periodontitis, the oral microbiome, and oral cancer. This theme highlighted the impact of microbial communities on disease progression, emphasizing the role of the oral microbiota in the development of oral cancer. The blue cluster investigated inflammation and the specific bacteria involved in oral carcinogenesis. The yellow cluster focused on identifying and understanding various risk factors associated with oral cancer. This theme aimed to aid in the development of preventive strategies by identifying factors that can increase susceptibility to oral cancer.

The bacterium *F. nucleatum* emerged in our keyword analysis as a prominent focus in the field of oral cancer. *F. nucleatum,* a Gram-negative anaerobic bacterium initially identified as a periodontal pathogen, has recently been implicated in colorectal cancer development. Studies have demonstrated that *F. nucleatum* can contribute to cancer progression through multiple mechanisms. It can activate cell proliferation, promote cell invasion, induce inflammation, and evade immune surveillance, all of which play crucial roles in carcinogenesis [54,55].

Another bacterial strain that stood out in our keyword analysis was *P. gingivalis*. This bacterium promotes cancer cell proliferation by reducing apoptosis through PI3K/Akt signaling. *P. gingivalis* can stimulate tumor growth and metastasis by inhibiting p53 expression [56]. When grown in association with cancer cells, *P. gingivalis* has been shown to improve the aggressiveness of oral cancer cells through epithelial–mesenchymal transition-like changes and the acquisition of stemness. Prolonged exposure to *P. gingivalis* also promotes the migration and invasiveness of cancer cells while providing resistance against chemotherapeutic agents [57].

Periodontitis, a persistent advanced inflammatory gingival disease caused by bacterial dysbiosis, is associated with an increased risk of cancer development [58–60]. Periodontitis develops as dental plaque accumulates, leading to the formation of periodontal pockets and tissue destruction. In particular, patients with periodontitis are reported to have a 2–5 times higher risk of acquiring cancer compared to healthy controls [61]. This elevated risk can be attributed to periodontal pathogens, which enable and maintain a constant chronic infection and systemic inflammatory response. Furthermore, these periodontal pathogens can affect specific intracellular pathways, leading to reduced expression of pro-apoptotic proteins, increased cell migration and invasion, and enhanced metastasis. Interestingly, pathogens such as *F. nucleatum* and *P. gingivalis* have also been

found to activate cell transformation. All these changes contribute to the maintenance of the chronic inflammatory process and induce carcinogenesis. At the cellular level, these alterations can induce permanent genetic changes in epithelial cells due to continuous exposure to cell metabolites, leading to abnormal cell divisions and, ultimately, the development of carcinoma [56]. Therefore, the association between periodontitis and cancer highlights the importance of addressing chronic inflammation and bacterial dysbiosis in oral health to potentially reduce the risk of cancer development in affected individuals [62,63].

The identification of these specific bacterial strains and their roles in the development of oral cancer underscores the complex interaction between bacteria and the progression of the disease. Understanding the mechanisms through which these bacteria influence oral cancer is crucial for the development of targeted therapeutic strategies and preventive interventions. By elucidating the contributions of *F. nucleatum, P. gingivalis*, and other bacteria involved, we can potentially uncover novel therapeutic targets and approaches to mitigate the impact of these bacterial strains on oral cancer.

This study uses bibliometric methods to explore the development of research on the influence of bacteria on oral cancer. This approach has provided us with a deep understanding of the diverse aspects of the research. However, it is important to acknowledge and address several limitations in our approach. First, we utilized only one database (WoS), which may have resulted in the exclusion of relevant articles from other sources. To mitigate this limitation, future studies should consider incorporating multiple databases to ensure a more comprehensive coverage of the literature. Secondly, this study focused exclusively on original articles, which may have resulted in the exclusion of highly cited review articles. While our intention was to capture the real trends and research hotspots, it is important to recognize that review articles provide valuable summaries and interpretations of the existing literature. Future studies could consider incorporating review articles to provide a more comprehensive understanding of the field. Lastly, although citation analysis has been used in this study, it should be noted that a combination of quality assessments would be beneficial to identify truly high-quality articles. Unlike systematic reviews and meta-analyses that selectively include high-standard studies, bibliometric analysis aims to provide a broad and comprehensive overview of a research area. Incorporating additional quality assessments can further enhance the evaluation of the included articles. Despite these limitations, this study offers valuable information on the advancements and evolving trends in research on bacterial influence on oral cancer over the past decade. By acknowledging these limitations, future studies can build on our findings and employ more robust methodologies to further explore this important research area. The knowledge gained from our study contributes to the appreciation of the advancement of scientific understanding in the field of oral cancer and offers a deeper understanding of the disease at a fundamental level. In general, our study underscores the critical need for continued research and exploration in the field of oral cancer and bacteria.

## 5. Conclusions

In general, these findings highlight the significant impact of the oral microbiota on the development and progression of oral cancer. Understanding the dynamics and interactions between the microbial communities and the host's immune response provides insight into the underlying mechanisms of carcinogenesis. Specific bacterial species such as *Fusobacterium, Prevotella, Porphyromonas,* and *Lactobacillus* or dysbiosis within the oral microbiota can serve as potential biomarkers for assessing cancer risk. Targeting the inflammatory bacteriome and modulating the oral microbiota may hold promise for the prevention and treatment of oral cancer. More research is warranted on the role of periodontal pathogens, such as *Fusobacterium* and *Porphyromonas*, in persistent infection and inflammation during carcinogenesis. However, to develop effective preventive and therapeutic strategies, more research is required to elucidate the precise mechanisms underlying these

interactions and explore potential therapeutic interventions in this field. In addition to studying pathogenic bacteria, exploring the role of relative benign bacteria that inhibit local inflammation or reduce the inflammatory potentiation of the entire bacteriome may also hold promise for preventing cancer development. Advancements in this area of research have the potential to revolutionize oral cancer management and improve overall health outcomes for affected individuals.

**Supplementary Materials:** The following supporting information can be downloaded at: https://www.mdpi.com/article/10.3390/curroncol30100650/s1, Figure S1. Co-occurrence network of bacterium-associated keywords of relevant publications, Figure S2. Co-authorship network of different affiliations.

**Author Contributions:** Conceptualization, S.-W.H., J.-J.Y., and Y.-Y.L.; methodology, S.-W.H. and Y.-Y.L.; software, J.-J.Y. and Y.-Y.L.; validation, S.-W.H. and J.-J.Y.; formal analysis, J.-J.Y.; investigation, S.-W.H. and J.-J.Y.; resources, Y.-Y.L.; data curation, S.-W.H., J.-J.Y., and Y.-Y.L.; writing—original draft preparation, Y.-Y.L.; writing—review and editing, S.-W.H.; visualization, S.-W.H., J.-J.Y., and Y.-Y.L.; supervision, Y.-Y.L.; project administration, Y.-Y.L. All authors have read and agreed to the published version of the manuscript.

**Funding:** This research received no external funding.

**Institutional Review Board Statement:** Not applicable.

**Informed Consent Statement:** Not applicable.

**Data Availability Statement:** Data are available upon reasonable request.

**Conflicts of Interest:** The authors declare no conflicts of interest.

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
