# Peer review of "Mapping the Scientific Landscape of Bacterial Influence on Oral Cancer: A Bibliometric Analysis of the Last Decade’s Medical Progress"

_curroncol, doi:10.3390/curroncol30100650_

Round 1

Reviewer 1 Report

This article deals with a relevant topic about the oral microbiome and cancer. It has a defined research question, and the authors describe their methods well.

However, the discussion sections need to be improved before its publication. I've included for you the attached detailed comments and suggestions. Please revise them

I will be honored to review the revised version of this manuscript.

Moderate editing of English language required

Reviewer 2 Report

Dear Authors,

The manuscript entitled "Mapping the Scientific Landscape of Bacterial Influence on Oral Cancer: A Bibliometric Analysis of the Last Decade's Medical Progress" uses bibliometric methods to explore the development of research on the influence of bacteria on oral cancer.

Mainly in the Introduction Chapter, there is some data that need revision. I recommend a minor revision of the article. 

The following points need correction:

Lines 34-35: The metastases of oral cancer involve and other major regions such as the lungs. Add the most important sites affected.

Lines 36:37 DO insert the reference for this survival rate and detail the number of patients included in it. "it was estimated that about 31.5% of patients with oral cancer cannot survive for more than 5 years after diagnosis, even after being actively treated".

Lines 47-48 need correction. References 6 and 7 emphasize the presence of a reduced bacterial diversity in head and neck cancer.

Figure 2. Detail this criterion: Excluded studies: irrelevant to the topic.

Best regards!

Reviewer 3 Report

The topic of the manuscript is the bibliometric analysis of the potential association between oral cancer and bacteria.

The title and the abstract of the article are informative. The Introduction briefly presents the issue of oral cancer. The section "Material and Methods" relatively precisely describes the chosen study design. The sections "Results" and “Discussion” are interestingly written, including the study limitations. The Conclusions could be more "take-home" messages.

Some following points must be clarified/corrected for the further processing of this article.

1.       Keywords should be enriched with the proper MeSH terms.

2.       The methodology lacks information about who included records in the analysis, how inaccuracies between researchers were handled, etc.

3.       The more recent references from 2023 should be supplemented in the Introduction or in the Discussion.

4.       The Conclusions could be more "take-home" messages.

5.       Also, I doubt whether the paper should be qualified as a review.

Round 2

Reviewer 1 Report

The authors have addressed all my comments and suggestions well; they need to revise the scientific names of bacteria at the genus level and always be in italics font.

Minor editing of English language required

Author Response

Reviewer 1 comment:

The authors have addressed all my comments and suggestions well; they need to revise the scientific names of bacteria at the genus level and always be in italics font.

Author's reply:

Thank you for pointing that out. We have revised accordingly throughout this manuscript.

Reviewer 3 Report

The Authors responded to the comments of all Reviewers, improving the manuscript. I have no further comments.

Author Response

Reviewer's comment:

The Authors responded to the comments of all Reviewersimproving the manuscriptI have no further comments.

Author's response:

Thank you for taking the time to review our manuscript and for your valuable feedback.